# Promising Role of Alkaloids in the Prevention and Treatment of Thyroid Cancer and Autoimmune Thyroid Disease: A Comprehensive Review of the Current Evidence

**DOI:** 10.3390/ijms25105395

**Published:** 2024-05-15

**Authors:** Giulia Di Dalmazi, Cesidio Giuliani, Ines Bucci, Marco Mascitti, Giorgio Napolitano

**Affiliations:** 1Center for Advanced Studies and Technology (CAST), University “G. d’Annunzio” of Chieti-Pescara, 66100 Chieti, Italy; cesidio.giuliani@unich.it (C.G.); ibucci@unich.it (I.B.); gnapol@unich.it (G.N.); 2Department of Medicine and Aging Science, University “G. d’Annunzio” of Chieti-Pescara, 66100 Chieti, Italy

**Keywords:** alkaloids, drug discovery, thyroid cancer cell lines, thyroiditis, cell proliferation, apoptosis, anti-cancer agent

## Abstract

Thyroid cancer (TC) and thyroid autoimmune disorders (AITD) are among the most common diseases in the general population, with higher incidence in women. Chronic inflammation and autoimmunity play a pivotal role in carcinogenesis. Some studies, indeed, have pointed out the presence of AITD as a risk factor for TC, although this issue remains controversial. Prevention of autoimmune disease and cancer is the ultimate goal for clinicians and scientists, but it is not always feasible. Thus, new treatments, that overcome the current barriers to prevention and treatment of TC and AITD are needed. Alkaloids are secondary plant metabolites endowed with several biological activities including anticancer and immunomodulatory properties. In this perspective, alkaloids may represent a promising source of prophylactic and therapeutic agents for TC and AITD. This review encompasses the current published literature on alkaloids effects on TC and AITD, with a specific focus on the pathways involved in TC and AITD development and progression.

## 1. Introduction

Alkaloids are naturally organic compounds containing at least one basic nitrogen atom. These compounds are found primarily in plants as secondary metabolites and are important for plants in the defense against herbivores and pathogenic organisms [1]. Alkaloids have been classified, based on their heterocyclic ring system and biosynthetic precursor, into several groups, including imidazole, indole, isoquinoline, piperidine, pyrrolidine, pyrrolizidine, quinoline, quinolizidines, steroid, tropane, and terpenoid alkaloids [2,3]. Alkaloids have been intensely investigated because of their biological activity and therapeutic potential. They possess, indeed, a broad spectrum of biological activities [4], including anti-inflammatory, anti-oxidant, anti-microbial, anticholinergic, analgesic, antiangiogenic [5], anti-cancer [6,7], and immunomodulatory properties [8]. Some alkaloids, such as Vinca alkaloids, Quinolines (camptothecin and its derivatives), and Taxanes (paclitaxel and docetaxel) have been used to treat cancer since the 1960s [9], beyond their traditional use. Currently, a large number of plant derivatives are under investigation for positioning and “repurposing” in cancer therapy [10].

Several reviews have examined the role of alkaloid in different types of cancer [7,9,11,12] and autoimmune diseases [8], but limited data are available on their effect on thyroid diseases. The effects of alkaloids on thyroid growth and function have been recently reviewed in [13], thus the goal of this review will be to discuss the potential role of alkaloids as prophylactic and therapeutic agents for thyroid cancer (TC) and autoimmune thyroid disease (AITD), focusing on the specific pathogenic pathways.

Worldwide, thyroid cancer is the most common endocrine malignancy and the fastest increasing cancer with wide geographical variation [14,15]. This significant rise in incidence rate is mainly attributed to the higher detection rate of differentiated thyroid cancers, particularly papillary thyroid cancer (PTC). Over the past 30 years, indeed, the incidence rate for follicular (FTC), anaplastic (ATC), and medullary (MTC) thyroid cancers have remained relatively stable [16]. The overall 5-year relative survival rates for thyroid cancer are 98.3 %. However, patients with poorly differentiated (PDTC) or anaplastic thyroid cancer (ATC) and those who are not responsive to the standard treatment of care (surgery and radioiodine according to the risk of recurrence) have a worse prognosis [17,18]. Furthermore, the radioactive iodine therapy is sometimes limited by the development of radioresistance and, thus, several strategies to overcome tumor radioresistance are currently under investigation [19]. These subsets of patients have limited treatment options and the highest unmet need for novel and effective antitumor molecules. In this view, plant alkaloids may represent an important source for drug discovery. Numerous alkaloids, indeed, have shown antiproliferative and anti-cancer effects on different types of cancers both in vitro and in vivo [6].

Similarly, autoimmune thyroid diseases, comprising Hashimoto’s thyroiditis (HT) and Graves’ Disease (GD), have a high prevalence in the general population (estimated around 5%) [20]. The prevalence may be even higher if considering the number of subjects with positive thyroid-specific antibodies without clinical evidence of thyroid dysfunction. Thus, agents able to inhibit or delay the autoimmune process with a favorable benefit–risk profile may have an extensive application.

For the specific aims of this review, a comprehensive literature search was conducted using various electronic databases such as PubMed, Google Scholar, and Scopus (search keywords were “alkaloids” and “thyroid” or “thyroid cancer” or “thyroid autoimmunity”). The emphasis of this search was on in vivo and in vitro animal or human studies investigating the effects of alkaloids on thyroid cancer and/or thyroid autoimmunity. Only English language full-text papers (time frame 1990–2023) were considered and included in this review. Irrelevant documents, review articles, incomplete articles, duplicates, and conference papers were excluded. References cited in the retrieved articles were also screened. A selection of relevant references related to the topic of interest was performed based first on title and abstract, and finally on the full text of the paper. Among 83 full texts that were examined, 41 were excluded and 42 were included in this review (as shown in Appendix A).

## 2. Results

### 2.1. Alkaloids and Thyroid Cancer

#### 2.1.1. Alkaloids Effects on Cell Proliferation and Cell Death

The majority of the studies available investigated the effects of alkaloids in vitro using representative cell lines for each type of thyroid cancer. Overall, alkaloids exert anti-cancer activities on human thyroid cancer cell lines through the induction of cell cycle arrest (Figure 1), apoptosis (Figure 2) and autophagy (Figure 3). Only few studies have been conducted using an in vivo model of thyroid cancer. Anti-cancer effects of alkaloids in vitro and in vivo are summarized in Table 1 and Table 2, respectively, and described below in detail.

*Aloperine* is a quinolizidine alkaloid isolated from *Sophora alopecuroides Linn*, a traditional Chinese herb, used for its antipyretic, anti-inflammatory, and analgesic properties [56]. Aloperine has been shown to suppress cell proliferation in five human thyroid cancer cell lines, including papillary thyroid carcinoma (PTC), follicular thyroid carcinoma (FTC), poorly differentiated thyroid carcinoma (PDTC), and ATC cell lines [21], using the cell counting kit-8 (CCK-8) for determination of cell viability, crystal violet staining for colony-formation, and flow-cytometry for cell cycle analysis. Aloperine exerted its antitumor activity through the promotion of caspase-dependent apoptosis (and not regulating the cell cycle as shown in colon cancer cell line [57]), as determined by flow cytometry (FC) using the Annexin V-FITC/PI staining assay, and by western blot analysis (WB) examining the expression of PARP, Bcl-xL, Bid, caspase-3, -8, and -9 in the cells incubated with aloperine [21]. The same group demonstrated that aloperine regulated autophagy in multidrug-resistant ATC (MDR ATC) and multidrug-resistant papillary thyroid carcinoma (MDR PTC) cell lines via MAPK and PI3K/AKT/mTOR signaling pathways [22]. The autophagic activity within cells was monitored by the expression of LC3-II (autophagosomal marker) by WB and the autophagosome formation detected using immunofluorescence (IF) staining for LC3-II [22].

*Berberine* is a natural isoquinoline alkaloid found in roots, rhizomes, and stem bark of many plants including Hydrastis canadensis (Goldenseal), Berberis aquifolium (Oregon grape), Berberis vulgaris (Bayberry), Coptis chinensis (Coptis), and Berberis aristata (Tree turmeric) [58]. Berberine has shown anti-cancer activity in several types of cancer, including thyroid cancer [59]. In particular, Berberine inhibited in a dose-dependent manner the growth of thyroid cancer cell lines TPC1 and 8505C through apoptosis assessed by Annexin V-FITC/PI staining and/or cell cycle arrest determined by FC using propidium iodide staining and p27 expression by WB [23]. Similarly, Li et al. [24] showed that berberine inhibited the proliferation of TPC1 cell lines and two ATC cell lines: the C643 (with H-RAS mutation) and the OCUT1 (with BRAFV600E and PIK3CA mutations) without affecting normal human thyroid cells (Htori3) [24]. The inhibitory effects of berberine were mediated by MAPK and PI3K/AKT/mTOR, which led to G0/G1 cell cycle arrest (assessed by FC) and mitochondrial apoptosis (determined by Annexin V-FITC staining) [24]. They also showed that berberine suppressed cell migration in a dose-dependent manner as demonstrated by the wound-healing assay and by the decreased expression of vimentin (by WB) after treatment [24]. Berberine was also tested in the human medullary thyroid carcinoma (MTC) cell line TT [26]. Authors showed that berberine suppressed the expression of the RET proto-oncogene at both mRNA and protein levels. The selective suppression of RET further inhibited the cell proliferation (arrest in G1), through the down-regulation of the PI3K/AKT pathway, which in turn activated apoptotic cell death (reduced Bcl-2 expression by WB and increase in the caspase-3 activity assessed by the ApoAlert Caspase Fluorescent Assay kit, Clonetech Laboratory, Mountain View, CA, USA) [26].

*Camptothecin*, a cytotoxic quinoline-based alkaloid extracted from the Chinese plant Camptotheca acuminata, is an inhibitor of DNA Topoisomerase-I with several anticancer properties, though its use is limited by its excessive toxicity, low stability and solubility [7]. To overcome this barrier, Gigliotti et al. loaded *Camptothecin* in cyclodextrin-nanosponges (CN-CPT) and tested this formulation on the ATC cell lines BHT-101 and CAL-62 [27]. CN-CPT inhibited the growth of both ATC cell lines (assessed by the MTT assay and by the clonogenic cell survival assay), arrested the cell cycle in the S phase (by FC using propidium iodide staining), and induced apoptosis (determined by an Annexin V staining kit, BD, Franklin Lakes, NJ, USA and by a fluorometric caspase-3 activity assay kit, MBL, Watertown, MA, USA) [27]. CN-CPT, also, inhibited tumor cell adhesion to endothelial cells, migration (as assessed by using a wound healing assay and a Boyden chamber assay) and secretion of pro-angiogenic factors (IL-8 and VEGF-α) [27]. Furthermore, Gigliotti et al. investigated the effect of CN-CPT in an orthotopic model of ATC obtained by the injection of Cal-62 cells into the thyroid lobe of female NSG mice, showing that CN-CPT significantly decreased the growth velocity, the development of the metastasis, and the tumor microvessel density (assessed by CD31 staining) of orthotopic ATC xenografts, without apparent toxicity [27]. These findings suggest that CPT nano-formulations may be a candidate in the treatment of ATC, generally resistant to standard therapy.

*Capsaicin*, a naturally occurring alkaloid found in the Capsicum family, has been extensively investigated as a potential anti-cancer agent in vivo and in vitro in several type of cancers; however, only one phase II clinical trial is currently ongoing in patients with prostate cancer [60]. In TC perspective, *Capsaicin* has been shown to inhibit cell migration and invasion of the metastatic PTC cells B-CPAP as assessed by the wound healing and transwell assay, and to downregulate the expression of the matrix metalloproteinases at both mRNA and protein levels (MMP-2 and MMP-9, involved in the epithelial–mesenchymal transition) [28]. Xu et al. demonstrated that *Capsaicin* exerted its inhibitory effects on BCPAP cells via the activation of the transient receptor potential vanilloid type 1 (TRPV1) channel; these effects were reversed by capsazepine, a competitive antagonist against the activation of TRPV1, as previously shown in other type of cancer [61]. The same group demonstrated that *Capsaicin* inhibited the growth of the ATC cells 8505C (assessed by an LDH release assay) and induced apoptosis via TRPV1 activation and subsequent mitochondrial calcium overload that led to mitochondrial dysfunction (as evidenced by the production of reactive oxygen species) [29].

*Colchicine* is produced from the Colchicum autumnale plant and is typically used to treat familial Mediterranean fever and gout flare-ups [32]. Colchicine has also certain inhibitory effects on hepatocellular carcinoma, prostate carcinoma, cervical cancer, and colon tumors [32]. Recently, colchicine has been shown to inhibit PTC cell proliferation and promote apoptosis due to its inhibitory effect on histone deacetylase 1 (HDAC1). The expression of HDAC1 was reduced at both mRNA and protein levels by colchicine treatment [32]. HDAC1, a molecule that regulates cell cycle progression and proliferation, has been characterized as a proteomic signature in thyroid cancer and its expression is linked to tumor size and progression [62].

*Cyclopamine*, a steroidal alkaloid, is a potent inhibitor of the sonic hedgehog pathway (SHH), which is highly activated in thyroid neoplasms and promotes thyroid cancer stem-like cell phenotype [33]. Cyclopamine has been shown to inhibit the proliferation of the FTC cell line WRO82 and the ATC cell lines KAT-18 and SW1736 by leading to cell cycle arrest or apoptosis (assessed by propidium iodide staining and FC) [33]. In similar studies, treatment with cyclopamine showed a time- and dose-dependent inhibition of the ATC cell lines C643 and Hth74 [35], and a decrease in cell motility and invasiveness of the ATC cell lines KAT-18 and SW1736, mediated by AKT and c-Met activation [34].

*Ellipticine* is a pyridocarbazole alkaloid isolated from Apocynaceae plants which exhibits antiproliferative activities against various tumor cells. Ellipticine and its derivatives NSC311152 and NSC311153 have been demonstrated to suppress the proliferation of the MTC TT cells by reducing the RET protein expression through the stabilization of the G-quadruplex structure formed within the RET promoter region [36]. Furthermore, the ellipticine derivative NSC311152 inhibited the tumor growth in an MTC xenograft mouse model, obtained through subcutaneous injection of the TT cells into SCID mice. The inhibition of tumor growth by NSC311152 in vivo was mediated by the reduced protein expression of RET, c-Myc, Bcl-2 and cyclin D1 in tumor tissues evaluated using western blotting [36].

*Evodiamine* is an indole quinazoline alkaloid extracted from the fruit of Tetradium ruticarpum, known as Evodia rutaecarpa (Wu-Zhu-Yu in Chinese), traditionally used in Chinese medicine to treat headache and gastrointestinal disorders [63]. Evodiamine has been shown to inhibit the growth of the ATC ARO cells by inducing G2/M cell cycle arrest (assessed by propidium iodide staining and FC) and caspase-dependent apoptosis (assessed by TUNEL assay and expression of caspases by WB) [37].

*Halofunginone* is an analog of febrifugine, an alkaloid originally isolated from the plant Dichroa febrifuga. It has been demonstrated to induce the amino acid response (AAR) and the autophagy pathway in the FTC WRO cells through proteasome degradation of the mammalian target of rapamycin (mTOR) [38].

*Harmine* is a β-carboline alkaloid isolated from Banisteriopsis caapi and Peganum harmala with antioxidant, anti-inflammatory, antitumor, anti-depressant, and anti-leishmanial activities [64]. Harmine has been proved to inhibit the growth of papillary thyroid cancer both in vitro and in vivo [39]. In vitro, harmine induced apoptosis of TPC-1 cells through regulating the ratio of Bcl-2/Bax and elevating the activity of caspase-3, and inhibited the migration and the invasion of TPC-1 cells in a dose-dependent manner (as detected by wound scratching assay and transwell assay) [39]. In vivo, harmine inhibited the growth of thyroid cancer in a xenograft model of TPC-1 cells in nude mice, without significant toxicity (such as weight loss or behavior) during the whole experiment [39].

*Indirubin*, a bis-indole alkaloid, is the active component of Danggui Longhui Wan, a mixture of plants (comprising Indigofera tinctoria and Isatis tinctoria) used in traditional Chinese medicine to treat chronic myelocytic leukemia. Indirubin derivatives, such as indirubin derivative 7-bromoindirubin-3′-oxime (7BIO), have been synthesized to improve the anti-tumor activity. 7BIO treatment showed inhibitory effects on 14 thyroid cancer cell lines (SW1736, HTh7, C643, HTh74, 8305C and 8505C as a model of ATC, BHT101, B-CPAP and TPC-1 modeling PTC, ML1, TT2609, FTC-133, FTC-236 and FTC238 modeling FTC) [40]. In particularly, 7BIO induced a non-classical caspase-independent cell death and DNA fragmentation in dedifferentiated and anaplastic thyroid carcinoma cells [40]. Similarly, 7BIO treatment caused an atypical cell death in the MTC TT cells with signs of necrosis and apoptosis induced by caspase-dependent as well as by caspase-independent pathways [41].

*Irinotecan* is a semisynthetic derivative of camptothecin, that has been used to treat various types of cancer [42]. Irinotecan encapsulated in nanoparticles to lower its systemic toxicity has been shown to induce cell apoptosis (by MTT assay) and reduce cell invasion (by wound healing assay) in the B-CPAP and FTC-133 cell lines [42]. Furthermore, irinotecan has been shown to inhibit the growth of the MTC cell line TT alone and in combination with the tyrosine kinase inhibitor, CEP-751 [43]. Irinotecan treatment was, also, effective against an MTC xenograft mouse model [43]. Similarly, the simultaneous combination of irinotecan and sunitinib (a multi-targeted receptor tyrosine kinase inhibitor) induced apoptosis in the ATC cells 8305C and FB3 cells, and decreased tumor growth in ATC xenografts [44]. In light of these encouraging results, a phase II clinical trial (NCT00100828) has been conducted to determine the response rate of irinotecan in patients with metastatic MTC. However, the study was closed to enrollment due to low accrual (six participants).

*Matrine* is a quinolizidine alkaloid extracted from the leguminous plant Sophora flavescens Ait, widely used in traditional Chinese medicine for treating various diseases. Cumulative data have demonstrated that matrine possess potent, broad-spectrum anti-cancer activities [56]. Matrine has been shown to induce apoptosis and G1 cell cycle arrest in the FTC cell line FTC-133 and in the PTC cell line TPC-1 by upregulating miR-21 and downregulating phosphorylated Akt [45,48]. In a similar study, matrine has been shown to induce the apoptosis of the PTC cell lines TPC-1, B-CPAP, and K1 in a dose-dependent manner, decreasing the level of Bcl-2 and activating caspase-3. Furthermore, it suppressed tumor growth in vivo in a PTC xenograft mouse model by downregulating the expression of miR-182-5p [46]. Recently, matrine has been reported to suppress proliferation, migration, and invasion of TPC-1 cells by up-regulating the expression of miR-192-5p, which is usually decreased in PTC tissues and cell lines [47].

*Oridonin*, a bioactive diterpenoid isolated from Rabdosia rubescens, has been shown to inhibit proliferation, migration and invasion of the PTC cell lines TPC-1 and B-CPAP [49]. Oridonin inhibited epithelial–mesenchymal transition and angiogenesis by downregulating the JAK2-STAT3 pathway [49]. The anti-tumorigenic effects of Oridonin were confirmed in a PTC xenograft mouse model. The tumor mass and volume, indeed, dramatically decreased in mice treated with Oridonin compared to that in vehicle-treated mice [49]. Furthermore, the expression of phosphorylated-JAK2, N-cadherin and VEGFA (by WB) was reduced in tumor tissues, indicating that Oridonin could suppress angiogenesis in vivo [49].

*Piperine*, a piperidine alkaloid isolated from the plant Piper nigrum, has been reported to reduce, in combination with the polyphenol curcumin, the PTC TPC-1 cells viability (by MTT assay), to arrest the cell cycle (analyzed by FC) and to decrease the expression of p21, p53 and β-catenin (by WB) [50].

*Piperlongumine*, also known as piplartine, is an amide alkaloid isolated from Long Pepper (*Piper longum*), and it has shown extensive biological activities, including anticancer properties [51]. Piperlongumine inhibited cell proliferation and colony formation of four human TC cell lines (IHH-4, WRO, 8505c, and KMH-2 cells) in a dosage-dependent manner [51]. Piperlongumine also promoted the intrinsic-caspase dependent pathway and the ROS-modulated Akt pathway of apoptosis [51]. Piperlongumine exerted, also, a significant anti-tumor activity in xenograft model of PTC, reducing tumor growth without toxicity [51].

*Sanguinarine*, a benzophenanthridine obtained from the root of Sanguinaria canadensis, significantly inhibited cell proliferation of the PTC cell line TPC-1 and B-CPAP cells in a dose- and time-dependent manner (by CCK-8 assay). It induced cell apoptosis (assessed by using annexinV and dead cell kit cell analyzer, Burlington, MA, USA) and autophagy (increased LC3 expression by WB) via inactivation of the signal transducer and activator of transcription 3 activation (STAT3) and by reactive oxygen species (ROS) production [52]

*Sinomenine* or *cocculine* is an alkaloid found in the root of the climbing plant Sinomenium acutum. It is known to possess anti-inflammatory, immunosuppressive, antitumor, neuroprotective, antiarrhythmic and other pharmacological effects [65]. Sinomenine has been shown to inhibit proliferation of B-CPAP and TPC-1 cells, induce cell apoptosis (assessed by Annexin V-FITC and propidium iodide dual staining) and cell cycle arrest (in G2/M phase) [53,54].

*Swainsonine*, an indolizidine alkaloid found in Astragalus plants (locoweeds), is a potent inhibitor of lysosomal α-mannosidase and Golgi mannosidase II. It causes severe toxicosis (locoism) in livestock grazing these plants. Nevertheless, swainsonine exhibits anti-tumor and immunomodulant activities. Swainsonine has been shown to modulate the activity of lymphokine-activated killer (LAK) cells against autologous thyroid cancer cells [55].

#### 2.1.2. Alkaloids Effects on Radioresistance and Chemosensitivity

Radioactive is a cornerstone in the treatment of patients with thyroid cancer, although it is sometimes limited by the development of radio-resistance. In this perspective, it is important to identify molecules that could act as radiosensitizer. Only two studies have investigated the role of alkaloids on radioresistance, focusing on the sodium/iodine symporter (NIS) that regulates the active transport of iodide into the thyroid gland. In particular, *Capsaicin* partially restored the sodium/iodine symporter (NIS) expression inducing a significant improvement of radioiodine avidity in ATC cells through the activation of the TRPV1 bypassing the TSH–TSHR pathway. [30]. Similarly, *Sinomenine* promoted protein expression and plasma membrane localization of NIS in the PTC cells B-CPAP and TPC-1 [53]. The upregulation of NIS, via the activation of the TSHR signaling pathway, induced an increased uptake of radioiodine [53]. Thus, both Capsaicin and Sinomenine could be potential therapeutic radiosensitizers. 

Chemotherapy (ChT) is generally used for metastatic ATC and rarely for advanced DTC, but it is associated with very low response rates and significant toxicities [16]. Thus, it is of great importance to identify a way to promote the chemosensitivity. Only one study has investigated the role of *Sanguinarine* in TC showing that this alkaloid was able to sensitize PTC cells to the chemotherapeutic drug cisplatin [52]. Currently data on alkaloids’ effect on chemosensitivity of ATC cells are not available. Further research should investigate whether alkaloids combined with conventional chemotherapy could enhance chemosensitivity of ATC cancer cells.

### 2.2. Alkaloids and Autoimmune Thyroid Diseases

Some alkaloids have been shown to possess immunomodulatory properties and, thus, a potential therapeutic implication in AITD (Table 3).

*Anatabine,* a minor alkaloid of Solanaceae plants, including tobacco, tomatoes, potatoes, peppers, and eggplants, has been shown to ameliorate experimental autoimmune thyroiditis (EAT) induced in 8-wk-old female CBA/J mice by thyroglobulin (Tg) injection [66]. In particular, anatabine decreased the antibody response to Tg, improved the recovery of thyroid function, and suppressed the EAT-mediated increase in IL-1 receptor type 2 and IL-18, suggesting a specific effect of anatabine on proinflammatory pathways [66]. Given the positive results observed in this preclinical model, a multicenter, double-blind, randomized, placebo-controlled clinical trial has been carried out to evaluate the effects of anatabine dietary supplementation in patients with HT [71]. HT patients had a significant reduction in absolute serum Tg antibodies (TgAb), but not thyroperoxidase antibodies (TPOAb), after 12 weeks of anatabine supplementation, confirming an immunological effect of anatabine in humans as shown previously in the mouse model of EAT [71].

*Cepharanthine*, a naturally occurring alkaloid extracted from the plant Stephania cepharantha Hayata, has been demonstrated to block T-cell activation by Tg peptides in non-obese diabetic (NOD) mice expressing DRb1-Arg74 (an HLA-DR variant conferring risk for AITD) immunized with human Tg [67]. Given these encouraging results, a dual effect polymeric system has been designed to treat, potentially, autoimmune hypothyroidism. The system involves the release of cepharanthine to block T cell activation and the release of selenium to decrease TPOAb levels [72]. Recently, cepharanthine has been revealed to block the binding of the TSH receptor peptide (TSHR.132) to HLA- DRβ1-Arg74 and, thus, to inhibit T-cell activation and cytokine response in a humanized, HLA-DR3 expressing, mouse model of Graves’ disease [68].

*Halofuginone*, an analog of the quinazolinone alkaloid febrifugine extracted from the plant Dichroa febrifuga, has been demonstrated to inhibit in vitro human T helper 17 cells (Th17), a distinct subset of CD4+ T cells involved in the pathogenesis of inflammatory and autoimmune diseases, by activating the amino acid starvation response (AAR) [73]. Halofuginone treatment has been shown to reduce the serum Tg Ab levels and to decrease the number of Th17 cells in the NOD.H-2h4 mouse model of spontaneous autoimmune thyroiditis [69].

*MYMD-1*, a synthetic tobacco alkaloid derivative, has been shown to decrease incidence and severity of autoimmune thyroiditis in NOD.H-2^h4^ mice acting on specific lymphoid subsets and on the cytokine response. In particular, MYMD-1 treatment reduced serum TgAb levels and dampened the subset of thyroidal CD3+CD4+Tbet+RORγT+ effector Th1 cells and the systemic levels of TNF-α, suggesting a potential clinical use of MYMD-1 as a novel immunometabolic regulator [70].

## 3. Conclusions

Alkaloids exhibit a promising role as anticancer and immunomodulatory agents in TC and AITD. By revising the preclinical studies currently available, we have showed that alkaloids regulate several pathways linked to cell proliferation, apoptosis and autophagy in human thyroid cancer cell lines. There are encouraging data that confirm the anti-tumorigenesis activity of alkaloids in xenograft models of thyroid cancer, laying the groundwork for human clinical trials. We also emphasized the role of alkaloids in damping down the autoimmune response in some murine models of thyroid autoimmunity. However, there are some limitations in the clinical use of alkaloids. The major challenge is the toxicity as some alkaloids may produce liver, kidney and neuron damage [74]. Well-conducted toxicity studies, focusing on the toxicity level and precise mechanism of action of alkaloids, are needed to overcome this limitation. Low solubility, weak stability, poor oral bioavailability, and short half-life are other factors restricting the clinical application of alkaloids. Nevertheless, most of these limitations can be reduced by applying novel drug delivery systems such as nanoparticles, liposomes, gels, and emulsions [75]. Although there is convincing experimental data, there are currently no human studies available on alkaloids in patients with TC. To date, indeed, only one randomized clinical trial has been conducted in AITD patients, showing an immunomodulatory effect of anatabine without any significant toxicity [71].

## Figures and Tables

**Figure 1 ijms-25-05395-f001:**
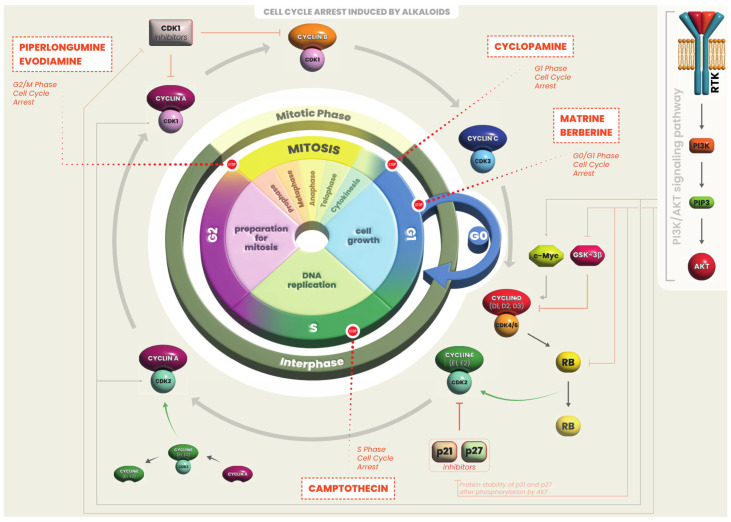
Regulation of cell cycle by alkaloids. The cell cycle is a four-stage (G1, S, G2, M) process that takes place in a cell leading to the production of two daughter cells. Cyclins and their catalytic partners, cyclin-dependent kinases (CDK), regulate this machinery. However, their regulation is uncontrolled in tumor cells, leading to abnormal cell growth. Alkaloids can arrest cell cycle at different phases, preventing thyroid cancer cells proliferation.

**Figure 2 ijms-25-05395-f002:**
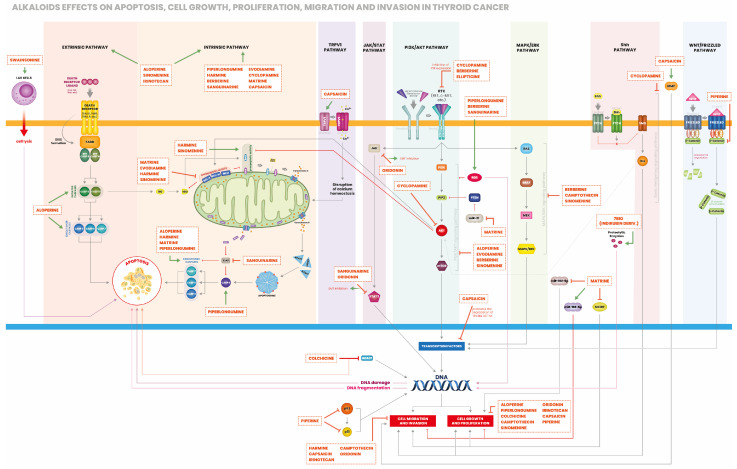
Regulation of thyroid cancer cell proliferation, apoptosis, migration and invasion by alkaloids. Uncontrolled proliferation and inhibited apoptosis are two major events for tumorigenesis. Therefore, inducing apoptosis and inhibiting cell proliferation are important strategies to prevent tumor progression. Alkaloids can promote apoptosis through activation of either the extrinsic (death receptor) or the intrinsic (mitochondrial) pathway. Alkaloids inhibit cell proliferation via several pathways: TRPV1, JAK-STAT, PI3K/AKT/mTOR, MAPK/ERK, SHH, WNT/frizzled as shown in the figure. Inhibiting cell migration and invasion is crucial to prevent metastasis at other sites. Some alkaloids inhibit cell migration and invasion by lowering matrix metalloproteinases expression, upregulating miR-192-5p (tumor suppressor) or inhibiting miR-182-5p.

**Figure 3 ijms-25-05395-f003:**
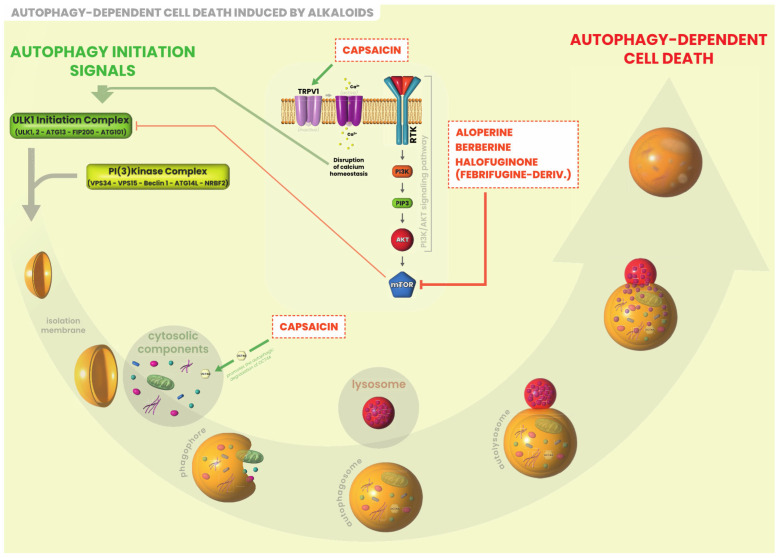
Alkaloids targeting autophagy in thyroid cancer. Autophagy, literally “self-digestion”, is essential for cell homeostasis. It suppresses tumor cell growth by degrading damaged mitochondria and proteins or inducing cell death. Aloperine, Berberine and Halofunginone promote autophagy via inhibition of and PI3K/AKT/mTOR signaling pathway leading to tumoral cell death. Capsaicin promotes the degradation of OCT4a, a cell stemness regulator.

**Table 1 ijms-25-05395-t001:** Anti-cancer effects of alkaloids in human thyroid cancer cell lines.

Alkaloids	Class of Alkaloids	Type of TC	Experimental Model	Anti-Cancer Effects	Ref.
*Aloperine*	Quinolizidine	FTC	WRO cells	- Inhibited the tumor cell growth - Induced caspase-dependent apoptosis	[21]
		PDTC	SW579 cells	- Inhibited the tumor cell growth - Induced caspase-dependent apoptosis	[21]
		ATC	8505C cells	- Inhibited the tumor cell growth - Induced caspase-dependent apoptosis	[21]
		MDR PTC	IHH-4 cells	- Promoted autophagy through the Akt/mTOR pathway suppression	[22]
		MDR ATC	KMH-2 cells	- Promoted autophagy through the Akt/mTOR pathway suppression	[22]
*Berberine*	Isoquinoline	PTC	TPC-1 cells	- Inhibited the tumor cell growth - Induced G0/G1cell cycle arrest - Upregulated p27 expression	[23]
		ATC	8505C, C643, OCUT1 cells	- Inhibited the tumor cell growth - Induced apoptosis - Induced apoptosis, G0/G1 cell cycle arrest and inhibited cell migration by modulating PI3K-AKT and MAPK pathways	[23] [24]
		ATC	CAL-62, BHT-101 cells	- Inhibited the tumor cell growth - Induced apoptosis and autophagy through the PI3k/Akt/mTOR pathway suppression and ROS generation	[25]
		MTC	TT cells	- Suppressed RET expression	[26]
*Camptothecin*	Quinoline	ATC	BHT-101, CAL-62 cells	- Inhibited cell proliferation, adhesion, and migration	[27]
*Capsaicin*	Capsacinoids	PTC	B-CPAP cells	- Inhibited cell migration and invasion via the activation of TRPV1 and downregulation of MMP expression	[28]
		ATC	8505C cells	- Induced mitochondrial dysfunction and apoptosis via the activation of TRPV1	[29]
		ATC	8505C, FRO cells	- Restored thyroid-specific gene expression and enhanced radioiodine uptake	[30]
		ATC	8505C, FRO cells	- Induced autophagy via the activation of TRPV1 - Promoted the degradation of OCT4A	[31]
*Colchicine*	Tropolone	PTC	TPC-1, KTC-1 cells	- Inhibited the tumor cell growth - Induced apoptosis via HDA1C inhibition	[32]
*Cyclopamine*	Steroidal alkaloid	FTC	WRO82 cells	- Induced apoptosis and cell cycle arrest	[33]
		ATC	KAT-18, SW1736 cells	- Induced apoptosis and cell cycle arrest - Decreased cell migration and invasion via AKT and c-Met activation	[33] [34]
		ATC	C643, Hth74 cells	- Inhibited the tumor cell growth	[35]
*Ellipticine*	Indole	MTC	TT cells	- Reduced RET expression	[36]
*Evodiamine*	Indole	ATC	ARO cells	- Induced cell cycle arrest and caspase-dependent apoptosis	[37]
Febrifugine and Halofunginone *	Quinazolinone	FTC	WRO cells	- Induced autophagy	[38]
*Harmine*	Carboline	PTC	TPC-1 cells	- Induced apoptosis by regulating Bcl-2/Bax and elevating the activity of caspase-3 - Inhibited cell migration and invasion	[39]
*Indirubin and 7BIO **	Indole	PTC	TPC-1, BHT101, B-CPAP cells	- Inhibited the tumor cell growth	[40]
		FTC	ML1, TT2609, FTC1-33, FTC-236, FTC-238 cells	- Inhibited the tumor cell growth	[40]
		ATC	SW1736, HTh7, C643, HTh74, 8305C, 8505C cells	- Inhibited the tumor cell growth - Induced a caspase-independent cell death and DNA fragmentation	[40]
		MTC	TT cells	- Induced apoptosis	[41]
*Irinotecan **	Quinoline	PTC	B-CPAP cells	- Induced cell death and reduced cell invasion	[42]
		FTC	FTC-133 cells	- Induced cell death and reduced cell invasion	[42]
		ATC	8305C, FB3 cells	- Inhibited the tumor cell growth	[43]
		MTC	TT cells	- Induced apoptosis	[44]
*Matrine*	Quinolizidine	PTC	TPC-1, B-CPAP, K1 cells	- Induced apoptosis and cell cycle arrest by up-regulating miR-21 and down-regulating phosphorylated Akt - Decreased the level of Bcl-2 and activated caspase-3 - Inhibited the tumor cell growth by downregulating the expression of miR-182-5p	[45,46]
		PTC	TPC-1 cells	- Inhibited cell proliferation, migration, and invasion via modulating miR-192-5p/SH3RF3 pathway	[47]
		FTC	FTC-133 cells	- Induced apoptosis and cell cycle arrest	[48]
*Oridonin*	Terpenoid	PTC	TPC-1, B-CPAP cells	- Inhibited cell proliferation, migration, and invasion - Inhibited angiogenesis and modulated EMT via the inactivation of JAK2/STAT3 signaling pathway	[49]
*Piperine*	Piperidine	PTC	TPC-1 cells	- Inhibited the tumor cell growth - Decreased the expression of p21, p53 and β-catenin	[50]
*Piperlongumine (or piplartine)*	Piperidine	PTC	IHH-4 cells	- Inhibited cell proliferation and colony formation - Promoted cell cycle arrest and cellular apoptosis through the intrinsic caspase-dependent pathway - Induced cellular apoptosis through the ROS-modulated Akt pathway	[51]
		FTC	WRO cells	- Inhibited cell proliferation and colony formation - Promoted cell cycle arrest and cellular apoptosis through the intrinsic caspase-dependent pathway - Induced cellular apoptosis through the ROS-modulated Akt pathway	[51]
		ATC	KMH-2, 8505c cells	- Inhibited cell proliferation and colony formation - Promoted cell cycle arrest and cellular apoptosis through the intrinsic caspase-dependent pathway - Induced cellular apoptosis through the ROS-modulated Akt pathway	[51]
*Sanguinarine*	Isoquinoline	PTC	TPC-1, B-CPAP cells	- Inhibited the tumor cell growth - Induced cell apoptosis via inactivation of STAT3 and reactive oxygen species generation - Sensitized PTC cells to chemotherapeutic drug cisplatin	[52]
*Sinomenine*	Isoquinoline	PTC	TPC-1, B-CPAP cells	- Inhibited cell proliferation - Induced cell apoptosis - Upregulated the membrane localization of NIS and enhanced radioiodine uptake	[53,54]
*Swainsonine*	Indolizidine	PTC	Autologous TC cells	- Modulated the activity of lymphokine-activated killer	[55]

Abbreviations: FTC, follicular thyroid carcinoma; PDTC, poorly differentiated thyroid cancer; ATC, anaplastic thyroid cancer; MDR, multidrug-resistant; PTC, papillary thyroid cancer; MTC, medullary thyroid cancer; Ref, references; 7BIO, indirubin derivative 7-bromoindirubin-3′-oxime; TRPV1, transient receptor potential vanilloid type 1; OCT4A, Octamer-binding transcription factor 4A; EMT, epithelial–mesenchymal transition; NIS, sodium/iodide symporter; MMP, matrix metalloproteinases; * synthetic derivative; STAT3, signal transducer and activator of transcription 3 activation; and TC, thyroid cancer.

**Table 2 ijms-25-05395-t002:** Anti-cancer effects of alkaloids on in vivo model of thyroid cancer.

Alkaloids	Class of Alkaloids	TC	Experimental Model	Effects	Ref.
*Camptothecin*	Quinoline	ATC	NSG mice with intrathyroidal injection of Cal-62 cells	- Decreased tumor growth, metastatization and tumor microvessel density	[27]
*Ellipticine derivative*	Pyridocarbazole	MTC	SCID mice injected s.c. with TT cells into the flank	- Decreased tumor growth - Reduced RET expression in tumor tissue	[36]
*Harmine*	β-carboline	PTC	Nude mice injected s.c. with TPC-1 cells in the left axillary space	- Decreased tumor growth	[39]
*Irinotecan*	Quinoline	MTC	Nude mice injected s.c. with TT cells into the flank	- Decreased tumor growth - Increased progression-free survival	[43]
		ATC	NU/NU nude mice injected s.c. with 8305C cells	- Decreased tumor growth - Increased caspase-3 activity in tumor tissue	[44]
*Matrine*	Quinolizidine	PTC	BALB/c nude mice injected s.c. with TPC-1 cells	- Decreased tumor growth - Downregulated the expression of miR-182-5p	[46]
*Oridonin*	Terpenoid	PTC	BALB/C nude injected s.c. with TPC-1 cells into the flank	- Decreased tumor growth - Reduced JAK-2, N-cadherin and VEGFA expression in tumor tissue	[49]
*Piperlongumine*	Piperidine	PTC	BALB/c nude mice injected s.c. with IHH-4 cells into the flank	- Decreased tumor growth	[51]

Abbreviations: ATC, anaplastic thyroid cancer; NSG, NOD scid gamma mouse; MTC, medullary thyroid cancer; PTC, papillary thyroid cancer; Ref, references; and TC, thyroid cancer.

**Table 3 ijms-25-05395-t003:** Immunomodulatory properties of alkaloids in autoimmune thyroid disorders.

Alkaloids	Class of Alkaloids	AITD	Experimental Model	Effects	Ref.
Anatabine	Nicotinic-acid derived alkaloids	AT	CBA/J mice Tg injected	- Decreased Tg-Ab levels - Suppressed the expression of IL-1 receptor type 2 and IL-18	[66]
Cepharanthine	Biscoclaurine	AT	NOD mice Tg injected	- Blocked T-cell activation	[67]
		GD	BALB/c-D3 mice injected with AdTSHR-289	- Inhibited T-cell activation and cytokine response	[68]
Halofuginone *	Quinazolinone	AT	NOD.H-2^h4^ mice	- Decreased Tg-Ab levels - Decreased Th17 cells	[69]
MYMD-1 *	Nicotinic-acid derived alkaloids	AT	NOD.H-2^h4^ mice	- Decreased Tg-Ab levels - Decreased Th1 cells and TNF-α levels	[70]

Abbreviations: AITD, autoimmune thyroid disease; AT, autoimmune thyroiditis; GD, Graves’ Disease; Tg-Ab, thyroglobulin antibodies; Tg, thyroglobulin; and * synthetic derivative.

## Data Availability

The data presented in this study are available on request from the corresponding author.

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
