# Peer review of "Promising Role of Alkaloids in the Prevention and Treatment of Thyroid Cancer and Autoimmune Thyroid Disease: A Comprehensive Review of the Current Evidence"

_ijms, 2024, doi:10.3390/ijms25105395_

Round 1
Reviewer 1 Report
Comments and Suggestions for Authors
Promising role of alkaloids in the prevention and treatment of thyroid cancer and autoimmune thyroid disease: a comprehensive review of the current evidence
Review work is important; however, authors must do an effort to strengthen it. Only a collection of quotes is observed, but the work lacks its own discussion. In this form it cannot be accepted for publication. In addition to the following observations:
The abstract is well done, but the keywords should be modified with words other than the title.
Line 42 -53. This section should be separated as a “Methodology Section”. And the introduction section (lines 29-41) should be greatly improved, giving emphasis to the problems of TC and AITD. Please add statistical data in this regard, at a local and international level. Use official databases such as the WHO, etc.
Review the entire manuscript and pay attention to the correct name of the species. All names are misspelled and must be italicized.
Figures 1 and 2. Clarify in the manuscript if the figures are your own authorship, in any case add the respective reference.
The Results section needs to be improved. The authors only make a collection of quotes but lacks discussion. A discussion section must be added, or in any case discuss in each of the sections.
Add a general description to each of the figures explaining the generalities observed.
The authors also do not mention any of their own opinions or what the perspectives are according to their review.
Author Response
Reviewer_1
Promising role of alkaloids in the prevention and treatment of thyroid cancer and autoimmune thyroid disease: a comprehensive review of the current evidence
- Review work is important; however, authors must do an effort to strengthen it. Only a collection of quotes is observed, but the work lacks its own discussion. In this form it cannot be accepted for publication.
- We thank the reviewer for this comment. We have done an extensively search of the published literature on this topic (alkaloids and thyroid) and to date this is the first review that have described the role of alkaloids in thyroid disease. While we acknowledge that the manuscript's text could be expanded, we believed that the figures and tables, developed by us with a rigorous scientific approach, based on a comprehensive review of the full texts of the included papers, provided a more elucidating presentation than textual description. For this reason, we have added a new figure showing the effect of alkaloids on autophagy (Figure 3) and a table (new Table 2) showing the effects of alkaloids on in vivo models of thyroid cancer. However, we have revised the entire manuscript accordingly to your suggestion. In particular, we have expanded and supplemented the following sections: introduction, results, and conclusion.
- The abstract is well done, but the keywords should be modified with words other than the title.
- We thank the reviewer for this comment. Updated keywords are highlighted in yellow.
- Line 42 -53. This section should be separated as a “Methodology Section”. And the introduction section (lines 29-41) should be greatly improved, giving emphasis to the problems of TC and AITD. Please add statistical data in this regard, at a local and international level.
- We thank the reviewer for this suggestion. Our review has been classified as a comprehensive narrative review, although we have done a meticulous search of the published literature on the field. The research strategy is showed in supplemental figure 1. Accordingly, to IJMS instructions for authors we had to delete the materials and methods section and we had to incorporate this section in the “Introduction section” (https://www.mdpi.com/journal/ijms/instructions#submission). This comment applies also to your request to add statistical data, our review is not a systematic review or a meta-analysis.
- Introduction section has been expanded with the epidemiology and social impact of thyroid cancer and autoimmune thyroid disease in the general population highlightening the potential role of alkaloids in their treatement (in yellow).
- Use official databases such as the WHO, etc.
- All names have been checked and revised accordingly.
- Review the entire manuscript and pay attention to the correct name of the species. All names are misspelled and must be italicized.
- All names have been checked and revised accordingly.
- Figures 1 and 2. Clarify in the manuscript if the figures are your own authorship, in any case add the respective reference.
- Figures 1 and 2 have been created by the authors considering the mechanisms of action summarized in tables 1, 2 and 3 after reading the original full text of every paper cited in the table. We spent a lot of times to draw these figures and to simplify the pathway regulated by alkaloids, a topic that “per se” could be demanding or tedious to read. Following your suggestion, we have added a figure legend.
- The Results section needs to be improved. The authors only make a collection of quotes but lacks discussion. A discussion section must be added, or in any case discuss in each of the sections.
- We have improved the results section adding a brief discussion of the results if needed.
- Add a general description to each of the figures explaining the generalities observed.
- We have added a brief “figure legend” to each figure.
- The authors also do not mention any of their own opinions or what the perspectives are according to their review.
- We have revised the conclusion section highlighting our perspectives on the actual limits of the use of alkaloid in clinical practice and the potential solutions to overcome these barriers.
Reviewer 2 Report
Comments and Suggestions for Authors
In the manuscript the role of alkaloids in the prevention and treatment of thyroid cancer and autoimmune thyroid disease have been reviewed. The subject matter of this manuscript is important. The search for new substances that could be used to treat various diseases, especially cancers and autoimmune diseases, for which effective therapies are still not always available is a very important scientific aspect. Study on the mechanism of action of candidates on drugs is particularly important.
The work is written correctly, the goal is clearly formulated, the results obtained by different authors have been presented and compared. Conclusions were concisely and correctly formulated. Possible future directions for further research have also been indicated.
However, in the description of many results regarding the activity of alkaloids, there is no short information about the used method of investigations. Such information can be very useful to many readers.
Author Response
In the manuscript the role of alkaloids in the prevention and treatment of thyroid cancer and autoimmune thyroid disease have been reviewed.
-The subject matter of this manuscript is important. The search for new substances that could be used to treat various diseases, especially cancers and autoimmune diseases, for which effective therapies are still not always available is a very important scientific aspect. Study on the mechanism of action of candidates on drugs is particularly important.
- The work is written correctly, the goal is clearly formulated, the results obtained by different authors have been presented and compared.
-Conclusions were concisely and correctly formulated. Possible future directions for further research have also been indicated.
- However, in the description of many results regarding the activity of alkaloids, there is no short information about the used method of investigations. Such information can be very useful to many readers.
We thank the reviewer for his/hers positive and constructive comments. We have revised the results section and added the principal methods of investigations that have been used by the researchers, focusing on the cell lines and the techniques used in the experiments. We have also added a table (new Table 2) showing the effects of alkaloids in xenograft models of thyroid cancer focusing on the experimental model that has been used and the principal results obtained.
Reviewer 3 Report
Comments and Suggestions for Authors
The alkaloids is a huge group of bioactive compounds with a broad range of potential actions. Annually, new scientific reports emerge concerning novel compounds or bioactive fractions of alkaloids, as well as their emerging applications in medicine. Therefore, it is so important to publish screening Rewiev about the latest data. The article “Promising role of alkaloids in the prevention and treatment of thyroid cancer and autoimmune thyroid disease: a comprehensive review of the current evidence” has a typical structure for a review article. The authors have prepared two highly significant and well-executed drawings on the subject of the regulation of the cell cycle by alkaloids and the regulation of thyroid cancer cells by alkaloids. The article comprises the most recent and most significant references, which have been carefully selected. The article has good overall merit, however, it requires some correction which can increase the quality of the article. Therefore, I recommended that a few changes be made, which are not obligatory but which have the potential to positively affect the reception of the article. Please find them below.
Comments to manuscript:
1. The introduction is somewhat brief. It would be beneficial to extend it and add a paragraph that outlines the historical discoveries of alkaloid drugs.
2. My suggestion is the addition of the figure with the structures of the main classes of alkaloids e.g. indolizidine, isoquinoline, piperidine, terpenoid etc. The relationship between structure and function will be easier to understand.
3. Figure 2 is a well-executed piece of work. However, the scale of the scheme is insufficient to permit the reading of all the details. It would be beneficial to alter the position to approximately 900 in a horizontal projection.
Comments to supplementary:
1. The number of records in the gap with “Records after duplicates removed” is missed. It should be supplemented.
Comments on the Quality of English LanguageThe content is at the level of an undergraduate course in English, with minor editorial and punctuation corrections.
Author Response
The alkaloids is a huge group of bioactive compounds with a broad range of potential actions. Annually, new scientific reports emerge concerning novel compounds or bioactive fractions of alkaloids, as well as their emerging applications in medicine. Therefore, it is so important to publish screening Rewiev about the latest data. The article “Promising role of alkaloids in the prevention and treatment of thyroid cancer and autoimmune thyroid disease: a comprehensive review of the current evidence” has a typical structure for a review article. The authors have prepared two highly significant and well-executed drawings on the subject of the regulation of the cell cycle by alkaloids and the regulation of thyroid cancer cells by alkaloids. The article comprises the most recent and most significant references, which have been carefully selected. The article has good overall merit, however, it requires some correction which can increase the quality of the article. Therefore, I recommended that a few changes be made, which are not obligatory but which have the potential to positively affect the reception of the article. Please find them below.
- We thank the reviewer for his/hers positive and constructive comments. We have improved the manuscript with the reviewers’ suggestions and we hope that the revised manuscript fulfilled the reviewers’ requests.
Comments to manuscript:
- The introduction is somewhat brief. It would be beneficial to extend it and add a paragraph that outlines the historical discoveries of alkaloid drugs.
- Introduction section has been expanded with the epidemiology and social impact of thyroid cancer and autoimmune thyroid disease in the general population highlighting the potential role of alkaloids in their treatment. We also added some historical notes on the discovery of key alkaloids.
- My suggestion is the addition of the figure with the structures of the main classes of alkaloids e.g. indolizidine, isoquinoline, piperidine, terpenoid etc. The relationship between structure and function will be easier to understand.
- A clear figure of alkaloids structure can be found in “DEY, Prasanta, et al. Analysis of alkaloids (indole alkaloids, isoquinoline alkaloids, tropane alkaloids). In: Recent advances in natural products analysis. Elsevier, 2020. p. 505-567”. Accordingly of what we have read there is no a striking link between the chemical structure and the biological activities of alkaloids. Indeed, the majority of alkaloids, independently of their structure, exerts anti-cancer activities regulating cell proliferation, apoptosis and autophagy. We have added a figure (Figure 3) showing the regulation of autophagy by alkaloids. If the reviewer believes that including a chemical structure figure would enhance the reader's understanding, we will prepare this figure without any issues.
- Figure 2 is a well-executed piece of work. However, the scale of the scheme is insufficient to permit the reading of all the details. It would be beneficial to alter the position to approximately 900 in a horizontal projection.
- We are grateful to the reviewer for this comment. We spent a lot of time and energy to draw this figure and to simplify the pathway regulated by alkaloids, a topic that “per se” could be demanding or tedious to read. We have moved figure 2 horizontally as per your request. We have also added a figure legend to simplify the reader.
Comments to supplementary:
- The number of records in the gap with “Records after duplicates removed” is missed. It should be supplemented.
- We have modified supplemental figure 1 accordingly (in yellow).
Round 2
Reviewer 1 Report
Comments and Suggestions for Authors The manuscript has been improved and can be accepted.Author Response
We thank the reviewer for his/her work on our manuscript. We are pleased that the reviewer appreciated our effort to satisfy his/her requests.